# Exploring the Effect of Water Content and Anion on the Pretreatment of Poplar with Three 1-Ethyl-3-methylimidazolium Ionic Liquids

**DOI:** 10.3390/molecules25102318

**Published:** 2020-05-15

**Authors:** Florence J. V. Gschwend, Jason P. Hallett, Agnieszka Brandt-Talbot

**Affiliations:** 1Department of Chemical Engineering, Imperial College London, London SW7 2AZ, UK; f.gschwend12@imperial.ac.uk (F.J.V.G.); j.hallett@imperial.ac.uk (J.P.H.); 2Department of Chemistry, Imperial College London, London W12 0BZ, UK

**Keywords:** ionic liquids, lignocellulose pretreatment, biorefining, enzymatic saccharification, compositional analysis, poplar

## Abstract

We report on the pretreatment of poplar wood with three different 1-ethyl-3-methylimidazolium ionic liquids, [EMim][OAc], [EMim][MeSO_3_], and [EMim][HSO_4_], at varying water contents from 0–40 wt% at 100 °C. The performance was evaluated by observing the lignin and hemicellulose removal, as well as enzymatic saccharification and lignin yield. The mechanism of pretreatment varied between the ionic liquids studied, with the hydrogen sulfate ionic liquid performing delignification and hemicellulose hydrolysis more effectively than the other solvents across the investigated water content range. The acetate ionic liquid produced superior glucose yield at low water contents, while the hydrogen sulfate ionic liquid performed better at higher water contents and produced a recoverable lignin. The methanesulfonate ionic liquid did not introduce significant fractionation or enhancement of saccharification yield under the conditions used. These findings help distinguish the roles of anion hydrogen bonding, solvent acidity, and water content on ionic liquid pretreatment and can aid with anion and water content selections for different applications.

## 1. Introduction

Glucose is a key substrate in biorefining, as it is readily fermented by a wide range of microbes into products, such as ethanol [1,2], butanol, sorbitol [3], and lactic acid [4]. Cellulose from low-cost wood resources is a promising source of sustainable glucose upon depolymerisation. Cellulose depolymerisation to glucose is possible using enzymes [5,6,7,8]; however, the enzymes can only access the cellulose contained in woody biomass after a pretreatment step. Various pretreatment methods are currently under development, some of which use ionic liquids (ILs) as solvents [9,10,11]. Cellulose-dissolving ILs have been found to be very effective in enabling fast and high-yielding sugar release from the pretreated material [12,13,14,15,16]. This has attracted a lot of interest in the research community, and a number of patents have been filed—for example, by Balensiefer et al. as one of the first ones, with a priority date of 2008 [17]. During dissolving IL pretreatment, woody biomass is dissolved or substantially swollen, and a regenerated pulp produced upon addition of an antisolvent. The ionic liquid is removed by washing with water. It is generally assumed that the presence of water negatively affects lignocellulose dissolution in these ILs [17,18], resulting in the need for energy-intensive drying of the biomass [12,16] and the IL [15]. However, Fu and Mazza have claimed that up to 50% water can result in the most digestible material after pretreatment of wheat straw with [EMim][OAc] using a 150–160 °C pretreatment temperature [19,20]. Yet, complete dissolution of a biomass is not crucial for the successful outcome of ionic liquid-based pretreatment. Certain IL-water mixtures have been reported as pretreatment solvents that achieve lignin and hemicellulose removal to increase the accessibility of the cellulose portion to enzymes, resulting in up to 100% of glucose release, albeit slower than after the dissolving pretreatment [21,22]. ILs with a high hydrogen bond basicity typically containing acetate [23] and chloride [24] anions have been reported to dissolve cellulose, while ILs with anions such as methyl sulfate and hydrogen sulfate have been reported to selectively dissolve lignin and hemicelluloses while the cellulose remains crystalline [22].

While much research has been carried out to optimise reaction conditions for both acetate [25,26] and hydrogen sulfate ILs [27,28], a direct comparison of the effect of increasing the water content for dissolving and lignin-extracting (known as ionoSolv) ILs with the same cation has not been published to date. Only Doherty et al. previously compared the effects of water on the pretreatment of another hardwood, maple wood, with [EMim][OAc], [BMim][OAc], and [BMim][MeSO_4_] ILs at 90 °C; however, the water range was limited to 0%–10% [18]. Brandt et al. compared [BMim][OAc], [BMim][MeSO_3_], and [BMim][HSO_4_] performance on *Miscanthus* at 120 °C for 22 h but only at 20 wt% water content [22].

In this study, we compare the performance of three ILs, 1-ethyl-3-methylimidazolium acetate [EMim][OAc], 1-ethyl-3-methylimidazolium methanesulfonate [EMim][MeSO_3_], and 1-ethyl-3-methylimidazolium hydrogen sulfate [EMim][HSO_4_], across a much wider water content range (0–40 wt%) for the pretreatment of milled poplar, using otherwise the same reaction and subsequent washing conditions. The ionic liquids were selected based on their hydrogen bond basicity and solvent acidity (Table 1). While the Kamlet-Taft β parameter (describing the solvent’s hydrogen bond basicity) for [EMim][OAc] is well above the cut-off required for cellulose dissolution (~0.8), the one for the methanesulfonate IL is just below the cut-off. The Kamlet-Taft hydrogen bond basicity is above the cut-off needed for lignin dissolution in all three cases (~0.40) [29]. 

This sheds a direct light on the interplay of the anion and water contents over a wide water content range and fully excludes any effects from the cation. We used the experimental procedures applied by Balensiefer et al., who also investigated the pretreatment of poplar [17]. We also probed Balensiefer’s claim that pretreatment with the water content of more than 15%, and preferably 10%, generally results in a less favourable pretreatment outcome.

## 2. Results

Poplar was pretreated with three ILs containing the 1-ethyl-3-methylimidazolium cation and three different anions (acetate, A; methylsulfonate, M; and hydrogen sulfate, H). The ILs contained between 0 to 40 wt% water, and samples are therefore named A0-A40, M0-M40, and H0-H40, to indicate the anion and water content used. Below, results on pulp appearance, yields, and composition are discussed, as well as glucose yields obtained during enzymatic saccharification and the yield of isolated lignin. The reaction conditions are based on the examples provided by Balensiefer et al. [17] We chose the water content range, as it contains previously investigated water content ranges by Doherty et al. [18] and Balensiefer et al. [17] for [EMim][OAc] but also extends to the water content ranges explored for [BMim][HSO_4_] [22]. The data from these studies taken together suggest that varying the water content across 0–40 wt% has the biggest effects on both cellulose dissolution (at the low end) and cellulose lignin separation (at the high end).

### 2.1. Pulp Appearance

Photographs of the pulps obtained after the pretreatment of poplar containing 0–40 wt% water at 100 °C for 46 h are displayed in Figure 1, with increasing water contents from left to right. Authors do not always show photos of the pulps; however, their appearances often give qualitative clues as to what has happened during the pretreatment.

The pulps produced with [EMim][OAc] at low water contents—A0, A5, and A10—had dinstinctappearances from each other and the remaining pulps (Figure 1a). Pulp A0 had a coarse yet soft consistency. Pulp A5 was swollen, elastic, and appeared moist even after drying. Pulp A10 was more similar in appearance to A0 but less dense. In contrast, pulps A15, A20, and A40 looked very similar to each other and were fine powders of light-brown colour. 

Figure 1b shows the pulps obtained after the pretreatment of poplar with [EMim][MeSO_3_]. All six pulps isolated with the methanesulfonate anion were fine powders with the same light colour; however, the particle size decreased noticeably as the water content increased.

As for the methanesulfonate pulps (Figure 1c), the hydrogen sulfate pulps consisted of fine particles, with the particle size decreasing with the increasing water content. The colour of the pulps were brown, similar to the colour of the A15-A40 pulps. The H pulps were the smallest in volume.

### 2.2. Pulp Yields

The pulp yields as a function of the water content for the three [EMim] ILs are displayed in Figure 2. Pulp yields were significantly affected by the water content when [EMim][OAc] was used under the conditions applied. The yield exceeded 100% for the 5% and 10% water content samples. This indicates that the ionic liquid was incorporated into the biomass under the studied conditions and this tallied well with the optical appearance of these pulps, which looked wet and had an increased volume (Figure 1a). Retention of excess IL was surprising, given that the pulp was washed with copious quantities of ethanol and water following the procedure outlined by Balensiefer et al. [17]. At water contents above 15%, pulp yields were around 80%, indicating a moderate amount of removal of biomass components by the acetate ionic liquid. 

Pulp yields were largely unaffected by the water content in the case of the methanesulfonate IL and were consistently between 93% and 97%, as shown in Figure 2, indicating very limited extraction of biomass components by this IL, which tallies well with their barely altered appearance. Pulp yields between 47% (H40) and 76% (H10) were obtained for the hydrogen sulfate IL, indicating the removal of certain biomass components. The pulp yield for H10 was unusually high, which we attribute to residual ionic liquid.

### 2.3. Compositional Analysis

Compositional analysis was carried out on the obtained pulps, as well as the raw biomass, and from this, the glucan recovery, hemicellulose removal, and lignin removal were calculated for the pretreated pulps. The results for all the ILs are also displayed in bar charts in the Appendix A. 

#### 2.3.1. Composition of [EMim][OAc] Pulps

In the case of [EMim][OAc] pretreated poplar, the compositional analysis shows that up to 30% of lignin and up to 61% of hemicellulose were removed (maximum lignin and hemicellulose removal at 20 wt% water was 29.8% and 61.4%, respectively; Figure 3). Apparent negative delignification and low glucan recoveries were found for 5 and 10 wt% water content (samples A5 and A10). This observation has been attributed to interference of the residual ionic liquid with the compositional analysis procedure, which relies on the hydrolysis of the structural carbohydrates using a defined acidity [32]. [EMim][OAc] is known as a basic ionic liquid due to acetic acid being a weak acid [30] and, hence, is able to lower the acidity in the compositional analysis solution. We therefore suspect that polysaccharides were not fully hydrolysed and remained as a solid instead, contributing to the gravimetrically determined acid insoluble lignin. Hence, compositional analysis data for the two samples are not reliable. The presence of residual ionic liquid was also indicated by the mass balances for compositional analysis (Appendix A). For the conditions where no significant incorporation of IL was observed, we found glucan/cellulose recovery above 90%.

#### 2.3.2. Composition of [EMim][MeSO_3_] Pulps

After the pretreatment of poplar with [EMim][MeSO_3_] containing 0–40 wt% water, over 90% of glucan—in most cases, around 95%—was recovered, while no more than 10.4% of lignin and 11.6% of hemicellulose was removed (Figure 4). The high glucan recovery and low hemicellulose and lignin removals confirm that the biomass composition was largely unaffected by pretreatment with this ionic liquid under the studied conditions, independent of the water content. The negative hemicellulose removal for M40 and lignin removal for M15 demonstrate the accuracy limitations of the compositional analysis method employed.

#### 2.3.3. Composition of [EMim][HSO_4_] Pulps

The results from the compositional analysis of [EMim][HSO_4_]-pretreated poplar are displayed in Figure 5. As indicated by the pulp yields (26%–53% of biomass extracted into IL), this IL removed the largest amount of biomass components. Indeed, we found that 72%–84% of hemicellulose and 54%–64% of lignin were extracted, while the glucan recovery was between 62% (H0) and 83% (H15). The [HSO_4_] IL is the only IL that resulted in substantial glucan loss. Pretreatment with water contents of 10% and lower resulted in the greatest glucan loss (glucan recovery below 75%), while the glucan recovery was stable at around 80% for water contents of 15%–40%. The H5, H10, and H15 samples appeared to contain residual ionic liquid based on the mass balances for the compositional analysis (Appendix A), similar to what was observed for [EMim][OAc]-pretreated poplar when the water contents was low. It appears that the residual [EMim][HSO_4_] affected the compositional analysis results less than residual [EMim][OAc], although the most contaminated sample, H10, seems to be an outlier for all three indicators, suggesting that IL contamination resulted in the underreporting of glucan and hemicellulose recovery and overreporting of the pulp lignin content.

### 2.4. Saccharification Yields

Enzymatic saccharification was carried out on the pretreated poplar as well as untreated poplar, and the results for 48 h of incubation are displayed in Figure 6. The highest saccharification yields were obtained with [EMim][OAc] at low water contents: A maximum yield of 90% (relative to the glucan content in the untreated poplar) was achieved with 5 wt% water present (A5), with a decline to below 50% in the presence of 20 wt% water (A20). The saccharification yield for A5 exceeded the measured glucan content in the pulp, as determined by compositional analysis (Figure 3). This further confirms the interference of residual IL with the compositional analysis method. While there was an issue with the compositional analysis, our data do not indicate that the saccharification yield was negatively affected by the presence of this ionic liquid in our experiments; this is discussed in more detail later on. 

After the pretreatment with [EMim][MeSO_3_], saccharification yields were not increased compared to the untreated biomass, irrespective of the water content in the ionic liquid (Figure 6), which is in agreement with the limited amount of extraction seen for this IL at all investigated water contents (Figure 2). It seems that the limited amount of particle size decrease had no positive effect on the enzymatic saccharification yield. 

After the pretreatment with [EMim][HSO_4_], saccharification yields were enhanced compared to the untreated biomass for all water contents and reached a maximum of 60% with 10 wt% water present. A further increase in the water content resulted in a gradual decline in saccharification yields to 41% for H40. For 20 wt% and 40 wt% water, [EMim][HSO_4_] slightly outperformed the acetate IL under the conditions applied. 

### 2.5. Lignin Yields

Lignin could be isolated after the pretreatment of poplar with [EMim][HSO_4_] but not for the [EMim][OAc] and [EMim][MeSO_3_] pretreated biomass, regardless of the water content in the ionic liquid. In the samples where lignin precipitated, it appeared after the water addition to the ethanol filtrate. It is known that water acts as an antisolvent for lignin dissolved in hydrogen sulfate ILs or in ethanol. The lignin was isolated via centrifugation, washed with water, and dried in vacuum. The yields upon drying are presented in Figure 7. We note that, although delignification was uniform across the investigated water contents for [EMim][HSO_4_] (Figure 5), the lignin yield exhibited a dependency on the water content, with nearly four times more lignin isolated at 15 wt% water content compared to 0 wt% water. Peak lignin yields were obtained at water contents of 10–20 wt%. This suggests that the water content has a substantial effect on lignin separation from the IL solution. Lignin yields were low, at below 5% of the initial biomass weight (less than 15% relative to the lignin content in the untreated biomass), which tallies well with the limited delignification observed (around 60%), although it is noteworthy that the lignin yield was lower than the delignification in all cases. 

## 3. Discussion

As mentioned earlier, the [EMim][OAc] treatment resulted in pulps with the highest digestibility of the cellulose portion, evidenced by glucose saccharification yields of up to 90% after pretreatment in the presence of 5 wt% water. This tallies well with previous findings, where a low water content was found to be best for [EMim][OAc] pretreatment. We note that Balensiefer et al. claim that water contents up to 15% result in high glucose release, while less than 10% is preferable [17]. Our results indicate that, under the conditions applied, there was indeed a 30% loss in saccharification yield at a 15 wt% water content compared to the optimum at a 5 wt% water content, confirming that a water content below 10% is desirable. 

However, our data for the 0–10 wt% range are somewhat different to what Doherty et al. found when pretreating maple wood with [EMim][OAc] for 12 h at 90 °C [18]. They reported a steady drop in saccharification yield as the water content increased, while we report a maximum at 5%. This could be due to the different reaction conditions (shorter time and lower temperature) and the different washing conditions. We show that only modest lignin and hemicellulose removal was observed with this IL, confirming that cellulose decrystallisation is a key enabler of increased saccharification yields, as observed by others [18,19]. 

We also note that Fu and Mazza found that a water content of 50% gave a higher overall sugar release than pure [EMim][OAc] [20]. However, their study used a much higher temperature (150 °C), and the optimum saccharification yield was determined for the total fermentable sugar yield, which included the hemicellulose. Similar optimum conditions for the overall sugar release (158 °C, 3.6 h, 49.5% water) were found in a subsequent design of experiment optimisation study carried out by the same authors. We note that cellulose digestibility alone was the highest for pure IL and at the lowest temperature investigated (130 °C). Shi et al. also investigated the effect of the water content on [EMim][OAc] pretreatment of switchgrass, using the conditions of 160 °C, 3 h and 0, 20, 50 and 80% water content [30]. Similar to Fu and Mazza, they found that glucan and hemicellulose recovery in the pulp were the highest at a 50% water content; however, enzymatic glucose and overall sugar release were best at 0% water content in [EMim][OAc]. It should be highlighted here that [EMim][OAc] is not stable at the optimum temperatures employed in the Fu and Shi studies [33]. We hence conclude that, while a wider range of water contents can be tolerated by [EMim][OAc] at high temperatures, at a temperature where this IL is stable (≤100 °C), a water content below 10% is best, as demonstrated in this study and other studies employing low temperatures.

Pulp yields exceeding 100% of the initial biomass weight indicate IL incorporation when using the washing conditions employed. Others have encountered difficulties removing [EMim][OAc] from regenerated pulp previously [26], and Liang et al. attributed it to the use of ethanol as the precipitation solvent [34]. The presence of [EMim][OAc] in the pretreated solid raises the question of whether the enzymatic saccharification could have been negatively affected. Similar concerns have been raised by Li et al., who observed decreasing saccharification yields from regenerated switchgrass with increasing quantities of residual [EMim][OAc] [25]. The highest saccharification yield (99.8% of the glucan in the pulp) was obtained at 3.5 wt% [EMim][OAc], which was the lowest concentration of [EMim][OAc] investigated. To compare with our study, we estimate the residual IL content in our experiments. Based on the pulp yields shown in Figure 2, we estimate that no more than 55 wt% (A5) and 26 wt% (A10) of the pulp samples were IL, and we assume a ~75% “ionic liquid-free” pulp yield as observed for A15. This would introduce a maximum of 55 mg (3.7 mmol) [EMim][OAc] into the 10 mL assay, which would equate to no more than 0.6 wt% IL present in the enzymatic saccharification. It should be noted that the assay also contained 0.5 mmol of citric acid acting as a pH buffer, which would have assisted with mitigating the pH changes brought about by the slightly alkaline IL. The mass balance for the compositional analysis procedures (Appendix A) indicates that 38% of the A10 pulp may have been IL. Hence, we conclude that, due to the comparatively low IL contamination and the presence of a buffer salt, the enzymatic saccharification yield was not affected in this study. This is supported by observing the highest glucose yield for the A10 sample despite the highest level of IL contamination. Nevertheless, optimisation of the coagulation and washing would be required, as high IL recovery is also a key aspect for the economic viability of IL-based pretreatment. The results further show that complete ionic liquid removal is important for an accurate compositional analysis, and compositional analysis mass balances can be used to monitor for significant IL contamination.

No appreciable effect on poplar was observed with [EMim][MeSO_3_] using the pretreatment conditions employed in this study, irrespective of the water content. This is interesting, as sulfonate ILs have been used successfully in biomass pretreatment before. A study from our group observed a significant pretreatment effect for the similar ionic liquid [BMim][MeSO_3_], including over a 90% glucose release during enzymatic saccharification; however, these experiments were carried out at 120 °C. This suggests that temperature may play an important role for activating methanesulfonate ionic liquids for pretreatment [22]. Tan et al. used 1-ethyl-3-methylimidazolium alkylbenzenesulfonate, [EMim][ABS], for the pretreatment of bagasse at 170–190 °C and reported substantial lignin extraction (and recovery) but did not perform enzymatic saccharification on the cellulose-rich pulps [35]. More investigation into sulfonate ILs is needed to understand the behaviour of this group of ILs across temperatures and water contents.

Pretreatment with [EMim][HSO_4_] resulted in significant hemicellulose and lignin extraction and a maximum observed glucose yield of 60% at 15 wt% water. There appeared to be residual ionic liquid in the samples pretreated with 0, 5, and 10 wt% water in the ionic liquid. We have previously observed a maximum saccharification yield of 85–90% at 10–40 wt% water for [BMim][HSO_4_] for *Miscanthus*; however, this was observed at a higher temperature of 120 °C and a shorter pretreatment time of 22 h [22]. We note that delignification was less extensive in this study than in others, and a strong correlation between the lignin removal and saccharification yield has been noted for hydrogen sulfate ionic liquids [36]. The fairly short saccharification time (48 h) employed here may have also augmented performance differences between [EMim][OAc] and [EMim][HSO_4_]. Short saccharification times are typically sufficient for [EMim][OAc] pretreated pulps due to the highly accessible, decrystallised cellulose, while sugar release from HSO_4_ IL-treated pulps is slower due to the high crystallinity of the cellulose in these pulps [37]. We think it is unlikely that the [EMim][HSO_4_] contamination affected the saccharification adversely due to the arguments made above for [EMim][OAc]. 

Overall, [EMim][OAc] seemed to be the most active IL at the lower temperature and short saccharification time applied here, while the other two ILs produce attractive glucose yields at higher temperatures and longer saccharification times. However, we note that more research is needed for sulfonate ILs than what is reported in the literature and here to conclude this with confidence. We note that the reaction time applied by Balensiefer et al. and reproduced in this study (46 h) is very long and would be disadvantageous for industrial applications. Shorter pretreatment times can be achieved at higher temperatures [36,38]; however, this may come at a cost of reduced ionic liquid recyclability in the case of [EMim][OAc] [33] and reduced cellulose stability in the case of hydrogen sulfate ILs [38]. 

Lignin recovery is an important and often underappreciated aspect of ionic liquid pretreatment. A lignin precipitate was isolated after employing [EMim][HSO_4_] upon the addition of water but not after employing [EMim][OAc], despite the latter also extracting some lignin. Low lignin yields without acidification have been observed before for [EMim][OAc] [22] and assigned to the acetate anion’s strong hydrogen bonding character, which keeps dissolved lignin in solution even in the presence of copious amounts of wash water. An effect of the water content on lignin precipitate yield has been previously reported for [BMim][HSO_4_] at 5%–80% water content at 120 °C [22]. Here, it is shown for [EMim][HSO_4_] and more resolved for the 0%–40% water range. It is currently unclear why the lignin yield varied significantly despite the lignin extraction being similar for all the tested water contents. We hypothesise that the differences in lignin yield could be due to an interplay of factors that impact the chemistry of lignin and hemicellulose extraction and conversion, such as availability of water and solvent acidity. It is hypothesised, for example, that the IL anion reacts with hydroxyl groups in the biomass to form sulfate esters at very low water contents. This hydrophilic modification could make the extracted lignin more water-soluble. For pretreatment at the higher water content, lignin fragments with higher water solubility may be extracted while more hydrophobic lignins remain with the cellulose, resulting in lower precipitation yields upon water addition. The formation of water-insoluble humins from hemicellulose may also affect precipitation yields. The effect of the water content on isolated lignin properties remains to be investigated.

## 4. Materials and Methods

### 4.1. Materials

All reagents were purchased from Sigma Aldrich (Gillingham, UK). [EMim][MeSO_3_], [EMim][HSO_4_], and [EMim][OAc] were purchased from Sigma Aldrich with a minimum purity of 95%. ^1^H, ^13^C, HSQC, HMQC, and HMBC NMR spectra were recorded on a Bruker 400 MHz NMR spectrometer. Chemical shifts (δ) were reported in ppm, the DMSO signal at 2.500 (^1^H dimension) and 39.520 (^13^C dimension). Mass spectrometry was measured by Dr Lisa Haigh (Imperial College, London, Chemistry Department, London, UK) on a Micromass Premier spectrometer (Waters, Milford, MA, USA). 

### 4.2. Feedstock

Untreated American poplar wood was obtained from W.L. West and Sons (Selham, UK). It was cut into pieces, chopped (RETSCH SM 2000), and sieved (RETSCH AS 200) to select a 180–300 µm size fraction (−50 + 80 US mesh scale) prior to use. It was air-dried and stored in plastic bags at room temperature in the dark.

### 4.3. Pretreatment

For pretreatment experiments, ca. 1 g of poplar (chopped and sieved to <300 µm) was weighed out into a 100-mL glass pressure tube (Ace Glass), and a total of 20 g of solvent added, the solvent consisting of ionic liquid with 0–40 wt% water. The tube was shaken in order to ensure wetting of the entirety of the biomass, and the tube then placed into a preheated fan-assisted oven at 100 °C. After 46 h at 100 °C, the tubes were removed from the oven and allowed to cool to room temperature. Eighty-seven millilitres of absolute ethanol was preheated to around 50 °C and the contents of the pretreatment tube transferred to a round bottom flask using the preheated ethanol to wash out the tube. The round bottom flask was placed on a hot plate and, under stirring, heated to 70 °C for 30 min. The round bottom flask was then removed from the heating plate and left to cool to room temperature. The suspension was vacuum-filtered using a Buchner filter and Whatman filter paper. The liquids were collected and the solids resuspended in 70 mL of distilled water in a round bottom flask with a stirrer bar. The round bottom flask was placed on a hot plate, and, under stirring, the suspension was brought to a boil for a few minutes. Then, the round bottom flask was removed from the heat and left to cool to room temperature. The suspension was vacuum-filtered using a Buchner filter and Whatman filter paper. The liquids were combined with the ethanol-IL filtrate obtained from the previous filtration. The combined filtrates were transferred to a round bottom flask, and under reduced pressure, the ethanol and water were removed. The solids were left to dry overnight and weighed the next morning. 

### 4.4. Feedstock and Pulp Characterisation

#### 4.4.1. Moisture Content

For the both native biomass and recovered pulps, the moisture content was determined according to the protocol “Determination of Total Solids in Biomass and Total Dissolved Solids in Liquid Process Samples” [39] published by the United States National Renewable Energy Laboratory (NREL), and details of how the procedure was carried out in our labs are described in the Appendix A. 

#### 4.4.2. Compositional Analysis

Compositional analysis was carried out according to the published procedure “Determination of Structural Carbohydrates and Lignin in Biomass” by the NREL [32]. Further details can be found in the Appendix A.

### 4.5. Enzymatic Saccharification

Enzymatic saccharification was carried out according to the published procedure “Low Solids Enzymatic Saccharification of Lignocellulosic Biomass” by the NREL [40]. Further details can be found in the Appendix A.

## 5. Conclusions

This study shows that both the ionic liquid anion and water contents together have key influences on enabling the enzymatic saccharification and lignin isolation after pretreatment with dialkylated imidazolium ionic liquids. Our results indicate that dissolving ionic liquids such as [EMim][OAc] are active at lower temperatures than fractionating ILs. In agreement with previous literature reports, they are most active at water contents below 15% when temperatures around 100 °Care applied. The lignin-extracting IL [EMim][HSO_4_] required a water content above 5% to sufficiently enhance enzymatic saccharification; it displayed a high tolerance to water above that limit. ILs with sulfonate anions seem to only be active above 100 °C, regardless of the water content, possibly due to lack of acidity or alkalinity which is needed to catalyse lignin depolymerisation and due to its medium-strength H-bond basicity, which prevents cellulose dissolution. Effective antisolvent isolation of lignin appears to be limited to ionic liquids that have acidic anions. We further show that ionic liquid contamination can reduce the accuracy of the compositional analysis procedure. This may be the case even when enzymatic saccharification is not affected. 

Overall, this study is evidence that ILs with hydrophilic anions can have very different optimum conditions and that the water content and temperature ranges for comparative investigations need to be chosen carefully. Furthermore, washing of the obtained pulp must be exhaustive to avoid low-quality analytical data, especially at lower water contents in the ionic liquid. Mass closures of the compositional analysis procedures can be used to monitor the amount of ionic liquid contamination.

## Figures and Tables

**Figure 1 molecules-25-02318-f001:**
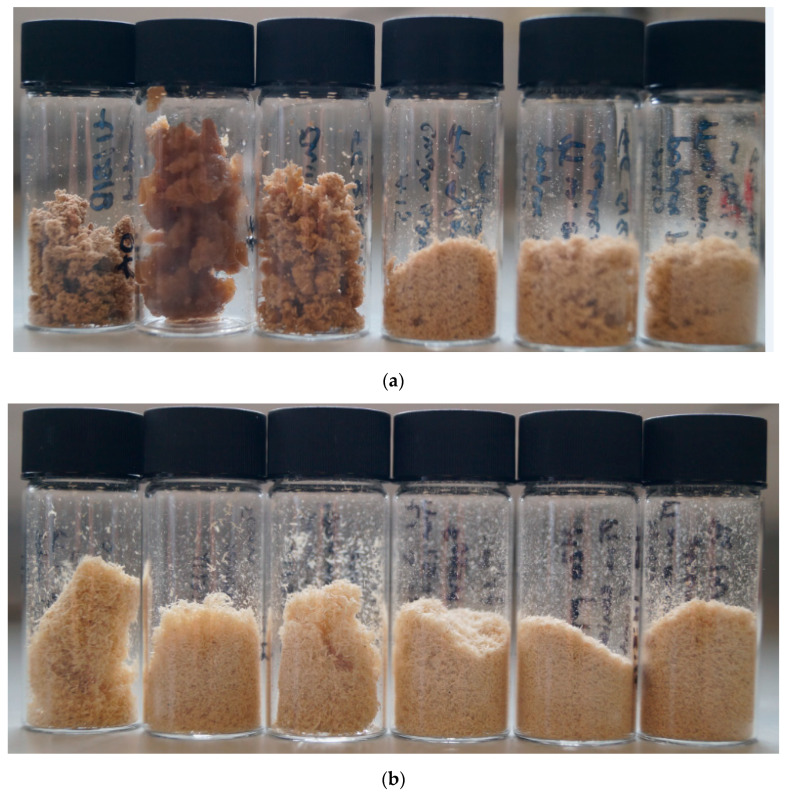
Poplar pulps isolated after pretreatment for 46 h at 100 °C with (**a**) [EMim][OAc], (**b**) [EMim][MeSO_3_], and (**c**) [EMim][HSO_4_] containing (from left to right) 0, 5, 10, 15, 20, and 40 wt% water.

**Figure 2 molecules-25-02318-f002:**
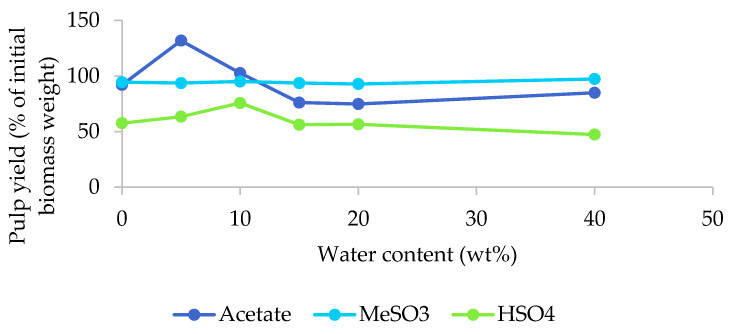
Pulp yields after the pretreatment of poplar with different [EMim] ionic liquids containing 0–40 wt% water for 46 h at 100 °C.

**Figure 3 molecules-25-02318-f003:**
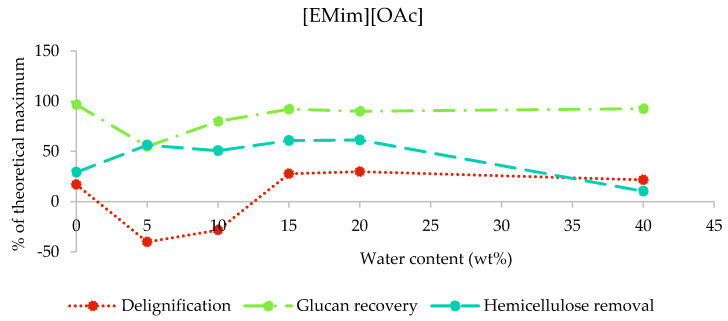
Delignification, glucan recovery, and hemicellulose removal after the pretreatment of poplar with [EMim][OAc] containing 0–40 wt% water for 46 h at 100 °C. The data for the 5 wt% and 10 wt% water samples are not reliable due to residual ionic liquid interfering with the compositional analysis procedure, as explained in Section 2.3.1.

**Figure 4 molecules-25-02318-f004:**
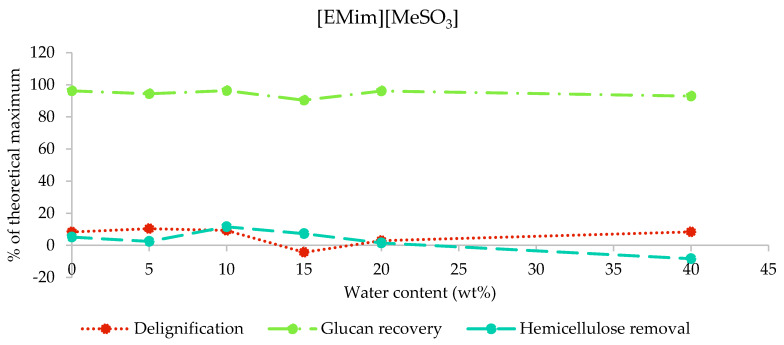
Delignification, glucan recovery, and hemicellulose removal after the pretreatment of poplar with [EMim][MeSO_3_] containing 0–40 wt% water for 46 h at 100 °C.

**Figure 5 molecules-25-02318-f005:**
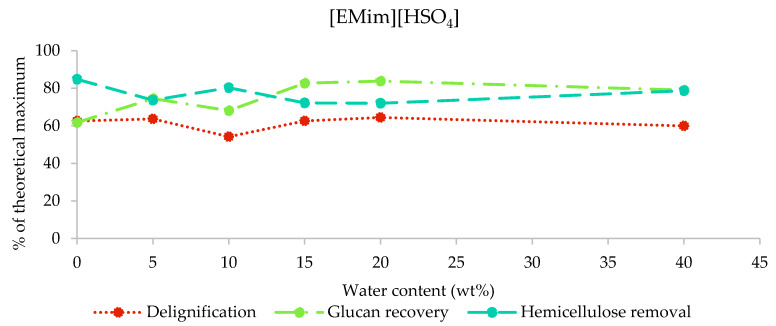
Delignification, glucan recovery, and hemicellulose removal after the pretreatment of poplar with [EMim][HSO_4_] containing 0–40 wt% water for 46 h at 100 °C.

**Figure 6 molecules-25-02318-f006:**
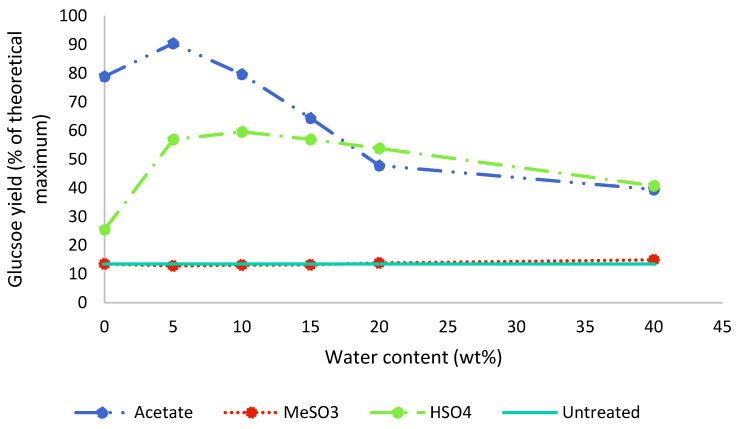
Enzymatic glucose release after 48 h of untreated and pretreated poplar. Poplar was pretreated for 46 h at 100 °C with different [EMim] ionic liquids containing 0–40 wt% water.

**Figure 7 molecules-25-02318-f007:**
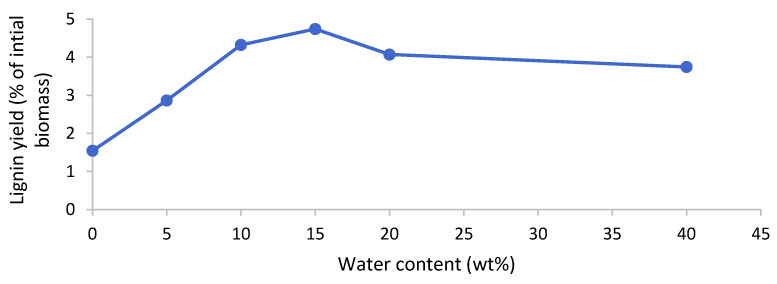
Lignin yield after pretreatment of the poplar with [EMim][HSO_4_] containing 0–40 wt% water for 46 h at 100 °C.

**Table 1 molecules-25-02318-t001:** Ionic liquids used in this study based and select solvent characteristics.

	Full Name	Hydrogen Bond Basicity (Kamlet-Taft β Parameter)	Solvent Acidity
**[EMim][OAc]**	1-ethyl-3-methylimidazolium acetate	1.23 [30]	Basic [30]
**[EMim][MeSO_3_]**	1-ethyl-3-methylimidazolium methanesulfonate	0.77 ^a^ [22]	Neutral
**[EMim][HSO_4_]**	1-ethyl-3-methylimidazolium hydrogen sulfate	0.67 ^b^ [22]	Acidic [31]

^a^ For 1-butyl-3-methylimidazolium methanesulfonate, and ^b^ for 1-butyl-3-methylimidazolium hydrogen sulfate.

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
