# Peer review of "Exploring the Effect of Water Content and Anion on the Pretreatment of Poplar with Three 1-Ethyl-3-methylimidazolium Ionic Liquids"

_molecules, 2020, doi:10.3390/molecules25102318_

Round 1
Reviewer 1 Report
The study is devoted to the water influence on the process of poplar pretreatment by EMim ionic liquids with different anions.
The manuscript is well written. There are two main disadvantages in the manuscript: (1) the study has been made in the field which was studied in details before, and therefore it is difficult to expect novel scientific results here; (2) all the results give the yields of glucose approximately 50 % based on the cellulose of the wood, and no more. Perhaps on these reasons, the reviewer did not see conclusions of high scientific and applied novelty in the Conclusions.
Several technical remarks:
- Figure 2 is not noted in the text.
- The sentence “We further show that ionic liquid contamination can reduce the accuracy of compositional analysis procedure, even when enzymatic saccharification is not affected” (lines 365-366) may be divided in two phrases.
- 255 Based on the pulp yields shown in 错误!未找到引用源。, we estimate that no more than 55 wt% (A5). 188 limited amount of extraction seen for this IL at all investigated water contents (错误!未找到引用源。 he pulp yields as a function of water content for the three [EMim] ILs are displayed in 错误!未113 找到引用源。.
Reviewer 2 Report
In the paper „Exploring the effect of water content and anion on 2 the pretreatment of poplar with three 1-ethyl-3-3 methylimidazolium ionic liquids” sent for review, the authors present the results of their research, in which they compare three ILs, 1-ethyl-3-methylimidazolium acetate [EMim][OAc], 1-ethyl- 3-methylimidazolium methanesulfonate [EMim][MeSO3], and 1-ethyl-3-methylimidazolium hydrogen sulfate [EMim][HSO4], containing 0 to 40 wt% water for the pretreatment of milled poplar using otherwise the same reaction and subsequent washing conditions.
In my opinion, the work may be published in its present state. However, minor corrections should be taken into consideration:
-line 113 in the sentence "The pulp yields as a function of water content for the three [EMim] ILs are displayed in 错误! 未 找到 引用 源" chinese letters should be changed to English,
-the same applies to lines 187 and 256,
-in reference [23], please add the page number, only 1326 is given, please add 1335, that is, it should be 1326-1335 as in the other items,
-in reference [17], a title written using only uppercase letters should be 'just like in a sentence', as in other literature items.
